Manuscript prepared for Ocean Sci.
with version 2014/09/16 7.15 Copernicus papers of the LaTeX class copernicus.cls.
Date: 14 June 2017

# The "shallow-waterness" of the wave climate in European coastal regions

Kai Håkon Christensen[1,3], Ana Carrasco[1], Jean-Raymond Bidlot[2], and
Øyvind Breivik[1,4]

[1]Norwegian Meteorological Institute, Henrik Mohns plass 1, N-0313 Oslo, Norway
[2]European Centre for Medium Range Weather Forecasts, Shinfield Park, Reading, RG2 9AX, UK
[3]Department of Geosciences, University of Oslo, Sem Sælands vei 1, N-0316, Oslo, Norway
[4]Geophysical Institute, University of Bergen, Allégaten 70, N-5007, Bergen, Norway

*Correspondence to:* Kai H. Christensen (kaihc@met.no)

**Abstract.** In contrast to deep water waves, shallow water waves are influenced by bottom topography, which has consequences for the propagation of wave energy as well as for the energy and momentum exchange between the waves and the mean flow. The ERA-Interim reanalysis is used to assess the fraction of wave energy associated with shallow water waves in coastal regions in Europe. We show maps of the distribution of this fraction as well as time series statistics from 8 selected stations. There is a strong seasonal dependence and high values are typically associated with winter storms, indicating that shallow water wave effects can occasionally be important even in the deeper parts of the shelf seas otherwise dominated by deep water waves.

## 1   Introduction

The purpose of this brief note is to present some aspects of ocean surface waves related to bottom topography. If the wavelength is small compared to the local water depth, the waves are unaffected by the presence of the sea floor and the wave energy balance is dominated by input from wind, dissipation by wave breaking and white capping, and nonlinear wave-wave interactions. If the wavelength is large compared to the local water depth, the situation is quite different and the wave energy propagation will directly depend on the bottom topography, with implications for dissipation and sediment transport in the bottom boundary layer, wave-mean flow interactions through wave radiation stresses, modification to the nonlinear wave-wave interactions, and so on (e.g., Komen et al., 1994; Smith, 2006).

The main aim of this study is to identify in which coastal regions in Europe shallow water wave effects may be important and to quantify the fraction of wave energy associated with ocean waves

that can "feel" the bottom. As such, this note differs from previous studies that focus on the wave climate, employing either hindcasts (e.g., Gorman et al., 2003; Dodet et al., 2010; Reistad et al., 2011; Aarnes et al., 2012), reanalyses (e.g., Dee et al., 2011; Reguero et al., 2012) or climate projections (e.g., Wang et al., 2004; Hemer et al., 2013) to assess average and/or extreme values of typical wave parameters on regional or global scales. Typical wave conditions can be classified according to the shape of the two-dimensional wave spectrum (e.g., Boukhanovsky et al., 2007), utilizing the fact that the waves will often be a combination of remotely forced swell and locally generated wind waves. In coastal regions, a significant proportion of the wave energy may be associated with waves on intermediate depth, and at any specific location this proportion will vary in time due to variations in the local and remote forcing of the waves. It should be emphasized that we do not make a clear distinction here between intermediate and shallow water waves, for which the wavelength is much larger than the local depth.

## 2 Concept and methods

We divide the wave spectrum into high and low frequency parts, using prescribed values of the ratio $n$ between the wave group and phase velocities to identify the frequency that separates the two parts. The wave energy in the low frequency part is divided by the total wave energy, and maps and time series statistics of this ratio are presented. Since wave dispersion depends on the local water depth in shallow waters, the frequency limit for any given $n$ will vary in space. The data are obtained from the wave model component of the ERA-Interim reanalysis (Dee et al., 2011).

### 2.1 Wave dispersion

The dispersion relation for surface gravity waves is

$$\omega^2 = gk \tanh kh. \tag{1}$$

Here $\omega$ is the wave angular frequency, $g$ is the acceleration due to gravity, $k$ is the wave number, and $h$ is the water depth. The phase velocity $c$ in the direction of wave propagation is $c = \omega/k$. The group velocity is given by $c_g = d\omega/dk$, and using (1) we have

$$n \equiv \frac{c_g}{c} = \frac{1}{2} + \frac{kh}{\sinh 2kh}. \tag{2}$$

The ratio $n$ between the group and the phase velocity is thus a function of the local water depth and the wave number. The limiting cases are for deep water ($kh \to \infty$), when $n = 1/2$, and for shallow water ($kh \to 0$), when $n = 1$ and the waves are non-dispersive. If $n > 1/2$, the waves are thus to some extent influenced by the bottom. In the present study we will consider $n$-values of 0.55, 0.65, 0.75 and 0.85. We will classify the waves according to their frequency $f = \omega/2\pi$, and for any given value of $n$ the corresponding frequency $f_n$ can be obtained from (1) and (2). To investigate the

"shallow-waterness" of a certain location we compute the ratio of energy $E_{sw}$ of the waves that feel the bottom to the total energy $E_{tot}$:

$$r_n = \frac{E_{sw}}{E_{tot}} = \frac{\int_0^{2\pi} \int_0^{f_n} F \, df \, d\theta}{\int_0^{2\pi} \int_0^{\infty} F \, df \, d\theta}, \tag{3}$$

where $F(f, \theta)$ is the directional wave spectrum obtained from the reanalysis data.

There are several options for the choice of parameter for the frequency cutoff. The ratio $n$ between the group and phase velocities occur naturally in radiation stress theory, which is the main reason why we use it here. A simple example of how (2) and (3) can be used is as follows: For monochromatic waves with energy $E$, the sum of the contribution to the radiation stress in the propagation direction from horizontal advection of momentum and the dynamical pressure below the mean (Eulerian) surface level is given by $2E(n - 1/2)$, which is zero for irrotational deep water waves (see Longuet-Higgins and Stewart 1964, and also Whitham 1962). The contribution from the divergence effect (e.g. Mcintyre, 1988) depends on the surface variance and yields an additional $E/2$. For any given $n$, the expression

$$\hat{S}_{xx} = r_n E_{tot}(2n - 1/2), \tag{4}$$

thus provides a lower bound (since $n$ increases with wavelength) for the radiation stress $\hat{S}_{xx}$ in the mean wave direction and should be suitable for assessing an order of magnitude estimate. A similar expression for the transverse radiation stress component can easily be derived. The net effect on e.g. the mean surface elevation will of course depend on the gradients in the radiation stresses and will vary from case to case.

### 2.2 ERA-Interim wave spectra

ERA-Interim (ERA-I) is a global coupled atmosphere-wave reanalysis starting in 1979 (Dee et al., 2011). An irregular latitude-longitude grid ensures relative constancy in atmospheric grid resolution towards the poles. T255 is the Gaussian grid with a spacing of the order 80 km, but atmospheric parameters are also made available (following bi-linear interpolation) on a $0.75 \times 0.75°$ regular latitude-longitude grid. The model and data assimilation scheme of the reanalysis are based on Cycle 31r2 of the Integrated Forecast System (IFS). The wave model WAM is coupled to the atmospheric part of the IFS through the exchange of the Charnock parameter. See Janssen (1989, 1991, 2004) for details of the coupling and Dee et al. (2011) for an overview of the ERA-Interim reanalysis. The resolution of the wave model model component is $1.0°$ on the Equator but the resolution is kept approximately constant globally through the use of a quasi-regular latitude-longitude grid where grid points are progressively removed toward the poles (Janssen, 2004). The spectral range from 0.035 to 0.55 Hz is spanned with 30 logarithmically spaced frequency bands. The angular resolution is $15°$ (24 bins). Full two-dimensional spectra are archived every six hours on the native grid. The ERA-I

| Name | Latitude | Longitude | Depth [m] | $H_s$ SI [%] | $H_s$ bias [m] | Collocation numbers |
|---|---|---|---|---|---|---|
| LF3J | 61.20 | 2.30 | 181 | 16.95 | 0.07 | 19395 |
| 62023 | 51.40 | -7.90 | 103 | 19.27 | 0.35 | 19400 |
| AUK | 56.39 | 2.05 | 79 | 15.03 | 0.02 | 2572 |
| 62069 | 48.29 | -4.97 | 63 | 19.53 | 0.22 | 6380 |
| LF5U | 56.50 | 3.21 | 60 | 14.49 | -0.07 | 27684 |
| K13 | 53.20 | 3.22 | 29 | 15.94 | -0.06 | 12910 |
| EURO | 51.99 | 3.27 | 28 | 17.77 | -0.09 | 12303 |
| BSH03 | 54.00 | 8.12 | 20 | 29.79 | -0.28 | 9113 |

**Table 1.** Station names, positions and depths, in addition to verification statistics for significant wave height ($H_s$): scatter index (SI, standard deviation of error divided by observation average) and bias. The rightmost column shows the number of collocated measurements used in deriving the statistics. The depths referred to here and in subsequent plots are the model depths.

WAM implementation incorporates shallow-water effects important in areas like the southern North Sea (Komen et al., 1994).

### 2.3 Stations

In addition to presenting maps of the ratio $r_n$, we analyse eight stations in some detail using the six-hourly time series from ERA-I. These stations correspond to locations with wave observations, and we have focused on the European Northwest Shelf Sea where shallow water waves are most prominent. The station names, positions, depths and some verification statistics are listed in Table 1, and the positions are also shown in Fig. 1. Three stations southwest of Ireland and the UK, and in the northern North Sea are exposed to long swell from the North Atlantic (62069, 62023, and LF3J), and all these stations are in intermediate to deep water (63 m, 103 m, and 181 m, respectively). Two stations are in intermediate depths in the middle of the North Sea (AUK and LF5U), while the rest are in the shallow southern part of the North Sea.

### 3 Results

We first investigate the spatial distribution of $n$. For this purpose we use monthly averages of the wave spectra. We then investigate the temporal variation of $n$ at the eight stations defined in Sec. 2.3, presenting monthly median values as well as the 5th, 25th, 75th and 95th percentiles. Finally, we plot $n$-values against mean period and significant wave height to investigate the variation of $n$ with wave steepness.

### 3.1 Spatial distribution

Values of $r_n$ are typically highest in the period December-March and Fig. 2 shows maps of the average values of $r_n$ for January for the period 1979-2012. Unsurprisingly, the highest values are found in shallow waters, including the North Sea, southwest of Ireland and UK, south of Spitsbergen, in the eastern part of the Barents Sea, and in the central Mediterranean. The monthly average $r_n$ ratios become, by necessity, smaller for increasing $n$, and is for $n = 0.85$ vanishingly small everywhere.

### 3.2 Seasonal dependence

Figures 3-4 show monthly values of significant wave height ($H_s$), mean period ($T_{m02}$), and $r_n$ for the eight stations listed in Table 1. The data are presented as median values and the 5th, 25th, 75th and 95th percentiles for the period 2003-2013. Significant wave height and mean periods are highest in the winter months, and the spread is also larger. The values of $r_n$ are quite small for the three

stations with largest depth, but we also see e.g. values of $r_{0.65}$ reaching 15% at station 62023 (103 m depth). Notably the $r_{0.65}$ values are lower for the shallower AUK station (79 m depth), which is explained by this station being sheltered from the long swell originating in the North Atlantic. The $r_n$ values are consistently lower in the summer months.

### 3.3 Dependence on wave steepness

Finally we investigate if high $r_n$ values are associated with a particular sea state, and Figs. 5-6 show scatter plots of all the data points in a $H_s/T_{m02}$ diagram. We only consider $n = 0.55$. For the deepest station LF3J there are only a few cases with relatively high $r_{0.55}$ values (to put this in context: there are over 16000 data points altogether). For the rest of the stations it is clear that $r_{0.55}$ is primarily correlated with the mean period, and not with the significant wave height, and high values can be

found both for high and low waves. There is a lower limit to the mean period that increases with the wave height, however, hence the average value of $r_{0.55}$ in general increases with $H_s$.

### 4   Conclusions

Data from the wave model component of the ERA-Interim reanalysis have been used to quantify the "shallow-waterness" of the wave climate in coastal regions in Europe. The "shallow-waterness"

is here defined as the ratio $r_n$ of wave energy of the components that are influenced by the bottom compared to the total wave energy. As can be expected, the ratios are largest during winter and on the European Northwest Shelf. Eight stations over that area have therefore been investigated in more detail.

    This work has a bearing on coupled wave-ocean modeling systems, for example, shallow water

wave-induced radiation stresses give rise to barotropic forcing terms that can play a role for storm surge modeling. The resolution of the ERA-Interim reanalysis is admittedly too coarse to provide

much detail in several regions such as the Baltic Sea and the Mediterranean subbasins. The point is, however, that a straightforward analysis of standard two-dimensional wave spectra from any wave model can provide some guidance on whether or not certain dynamical processes related to the "shallow-waterness" are important. All the necessary information to evaluate (4) can essentially be shown in scatter plots like Figs. 5 and 6.

Similar methods as we present here could also be used to investigate other processes, for example depth-refraction, although the cutoff criterion should in this case be based on the ratio between deep water and local wave number values (e.g. Holthuijsen, 2007). In addition, the influence of currents on the waves could be included using relative instead of absolute frequencies, which is likely to play a role in places with strong tidal flows such as at station 62069 (Ardhuin et al., 2012).

With the exception of the shallowest parts of the shelf seas, the "shallow-waterness" is on average quite small, but occasional high values of $r_n$ can be found in intermediate water depths ($\sim 100$ m). Destructive storm surge events are typically caused by intense winter storms with high waves, and our results suggest that in such situations shallow water effects can be important even at great distances from the coast. The "shallow-waterness" is primarily correlated with the mean period and can be found for both high and low waves, but shallow water effects become increasingly important for higher waves since these are associated with longer mean periods.

*Acknowledgements.* This work was funded by the European Union Seventh Framework Program FP7/2007-2013 under grant no 283367 (MyOcean2). Ø.B. was partly supported by the European Union FP7 project MyWave (grant no 284455) and in part by the ExWaMar project (grant no 256466) funded by the Research Council of Norway. The authors also want to thank Dr. Fabrice Ardhuin for providing corrections and constructive comments to an earlier version of this manuscript.

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

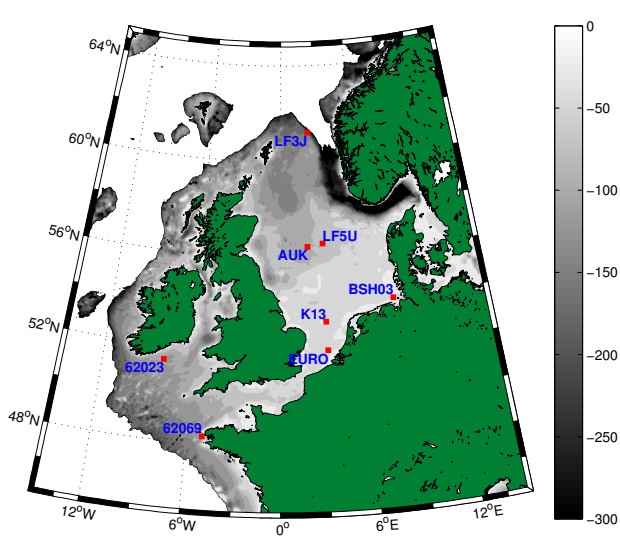

**Figure 1.** Map of station positions. Depths less than 300 meters are indicated in gray.

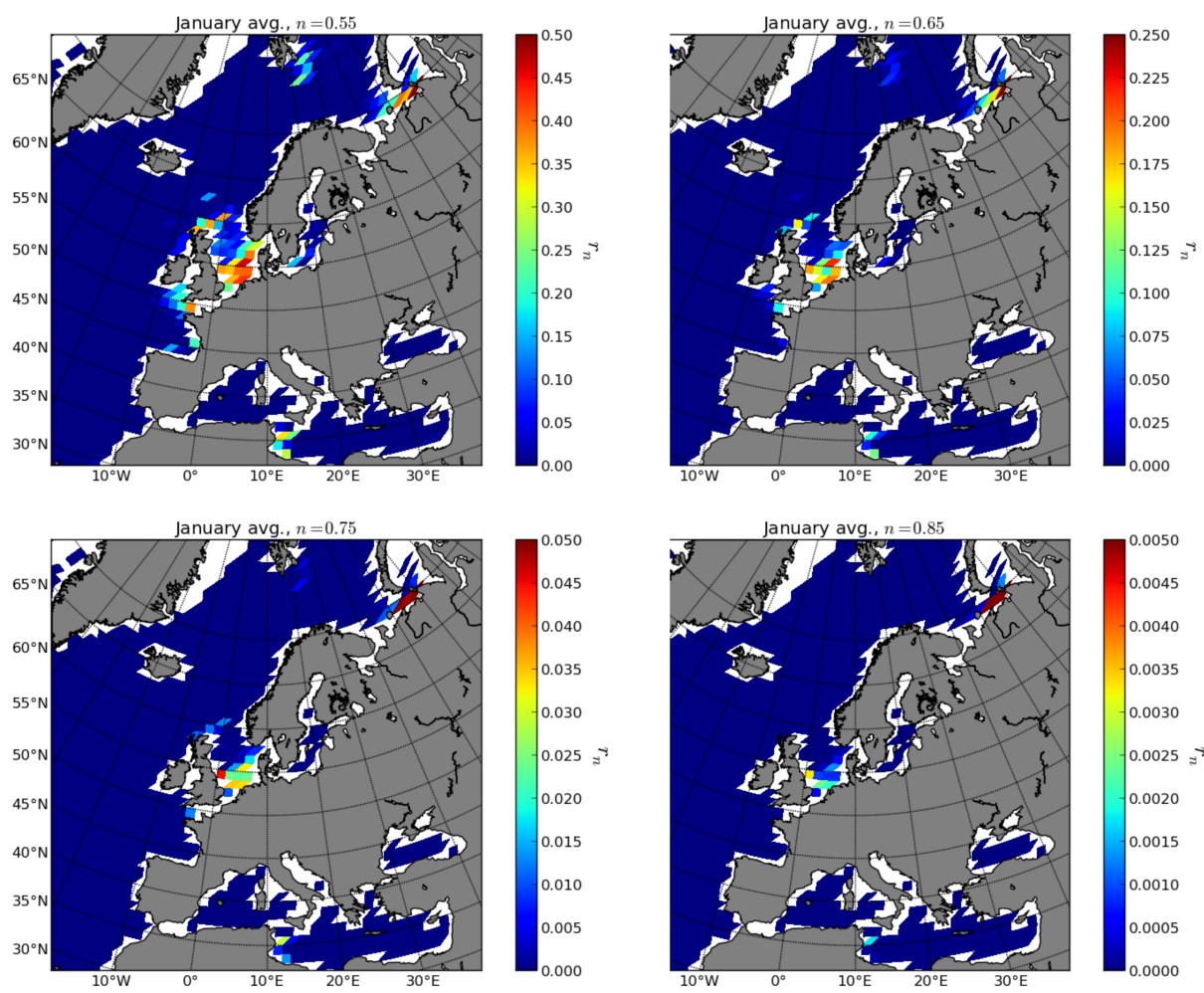

**Figure 2.** Average values of $r_{0.55}, r_{0.65}, r_{0.75}, r_{0.85}$ in January for the period 1979-2012.

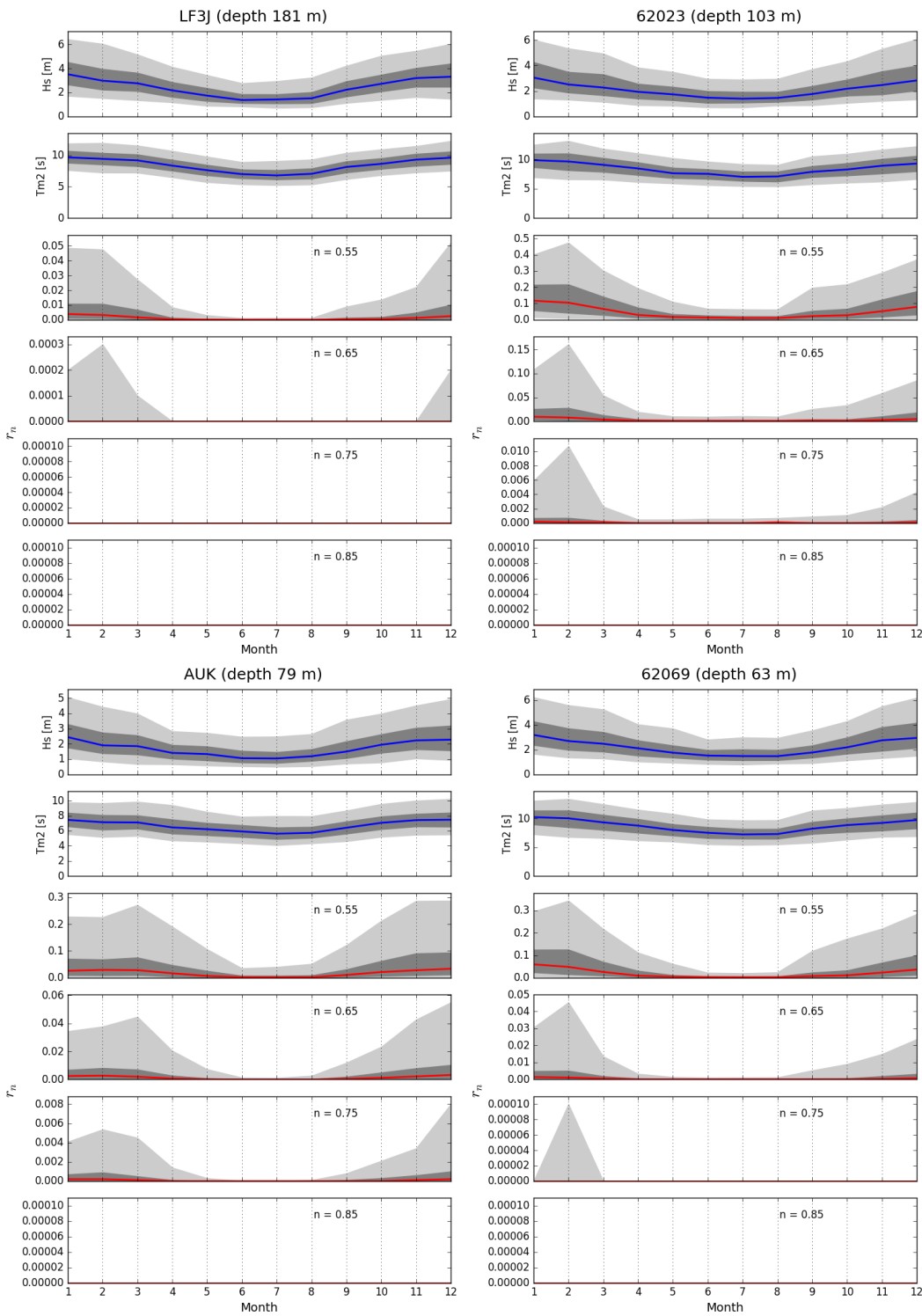

**Figure 3.** Monthly values of significant wave height, mean period, and $r_n$ values for $n = 0.55, 0.65, 0.75$ and $0.85$ at stations LF3J, 62023, AUK and 62069. Median values are given by red and blue lines; 25th to 75th percentiles are shown as dark gray; 5th to 95th percentiles are shown as ligth gray.

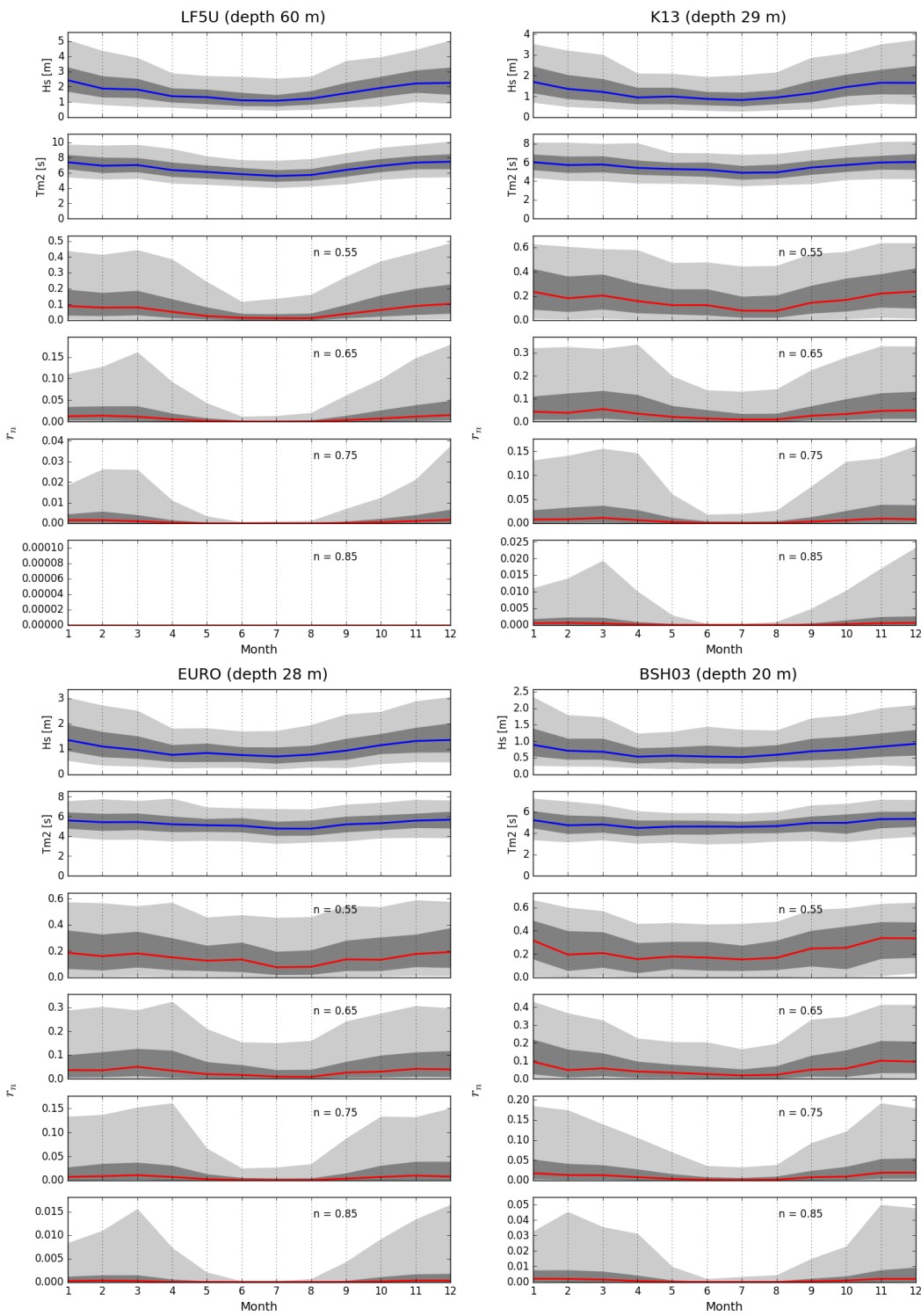

**Figure 4.** Same as Fig. 3, but for stations LF5U, K13, EURO and BSH03.

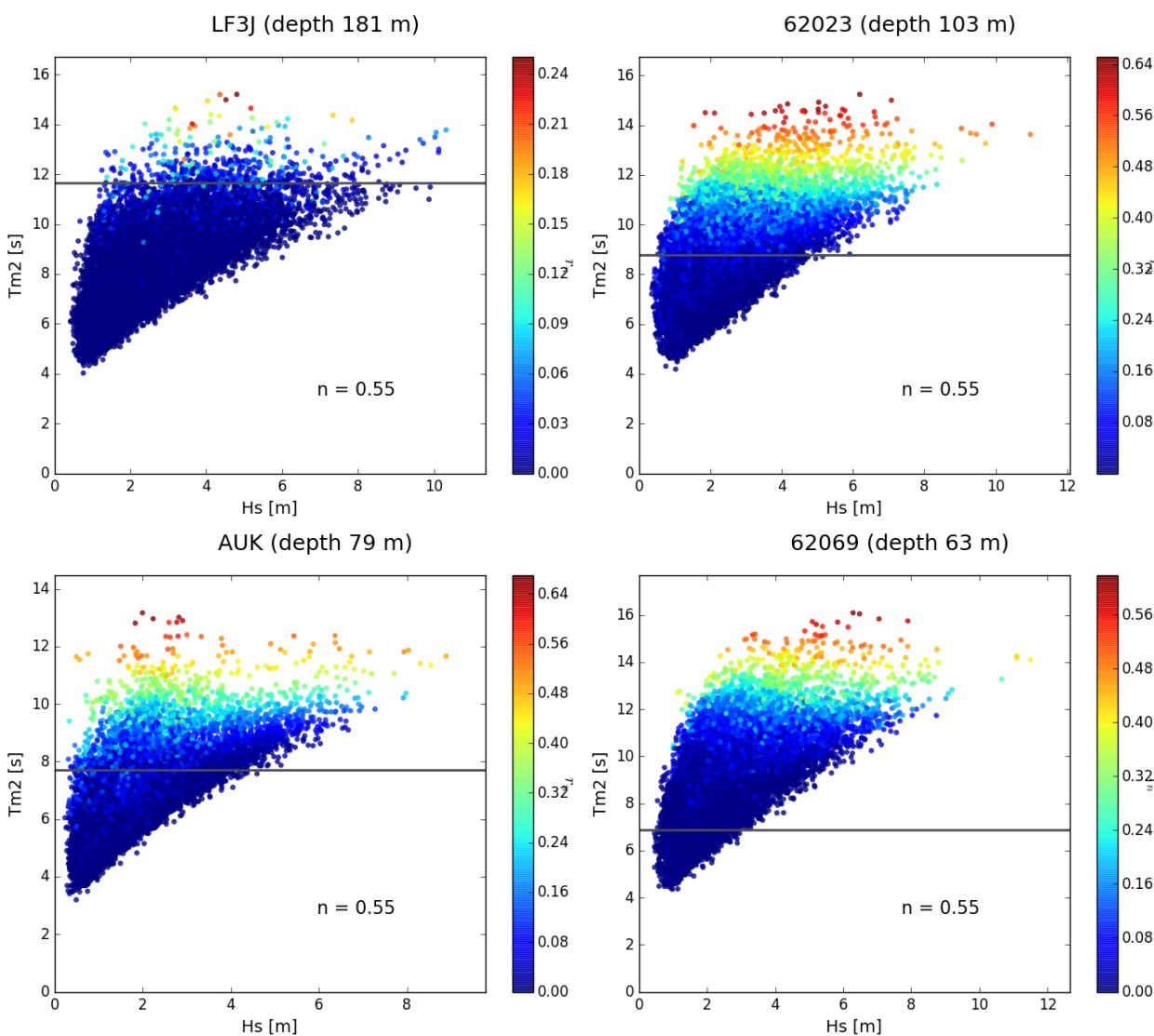

**Figure 5.** Scatter plot of all the data points for stations LF3J, 62023, AUK and 62069, with colors indicating $r_{0.55}$ values. The gray line indicates the period corresponding to $n = 0.55$ for each station.

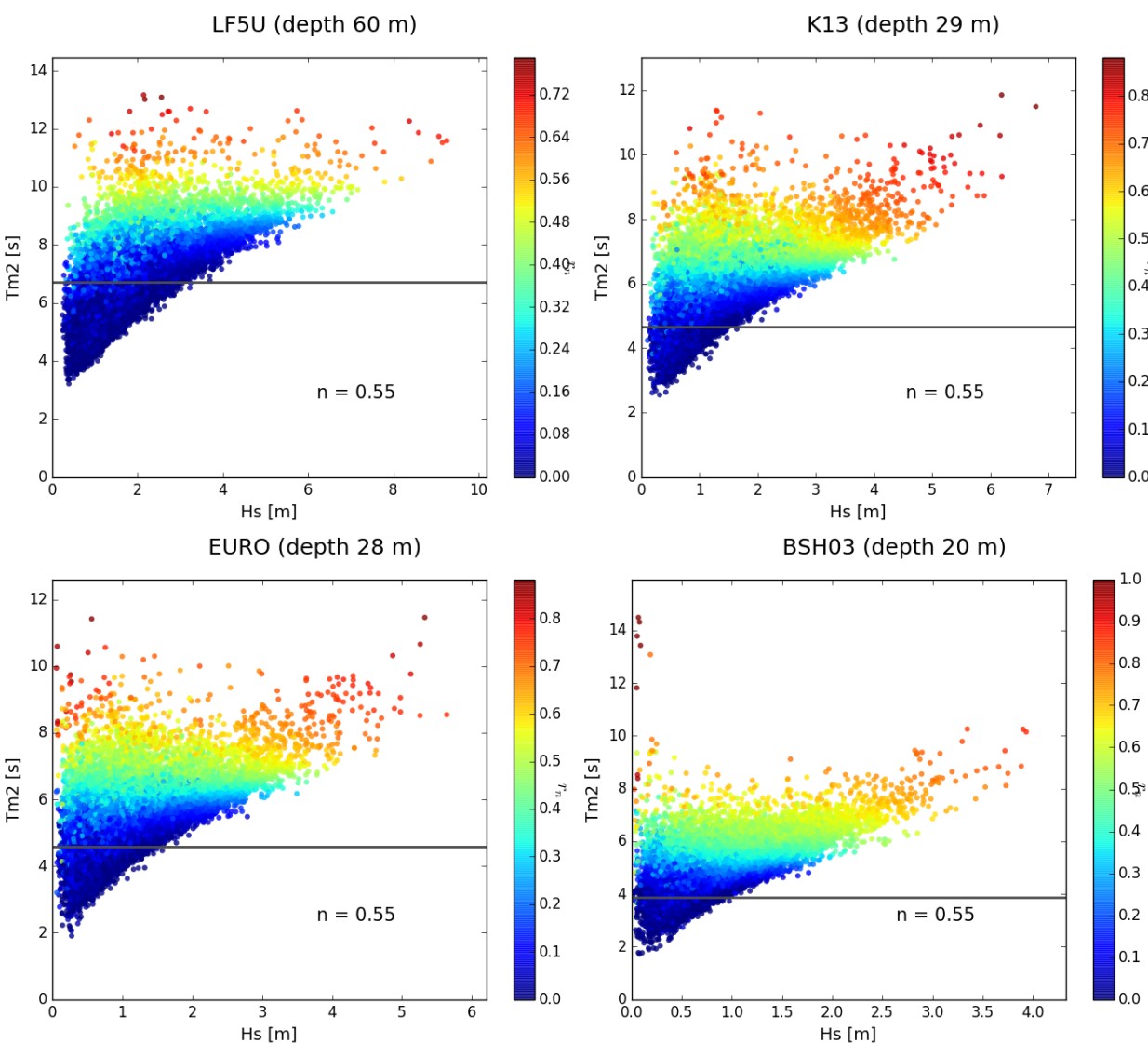

**Figure 6.** As Fig. 5, but for stations LF5U, K13, EURO and BSH03.