# Peer review of "The "shallow-waterness" of the wave climate in European coastal regions"

_Ocean Science, 2016_

## Referee Comment (RC1) · Anonymous Referee #1 · 27 Feb 2017

In this paper authors identify the fraction of energy that is affected by interaction with the bottom. They found that it depends on the mean wavelength (I would say this is obvious), it can be large also far from the coast if the water is sufficiently shallow (which in practice is the case for the central areas of the North Sea). Even where water is deep (100m), shallow water effects can be occasionally present if waves are sufficiently long.

While the paper is well written, concise and methodologically clear (I mean that mathematical definition -see formulas 2 and 3- of $r_n$ is clear, I have some difficulty to identify the real utility of this study. In my view authors should explain the practical relevance of a specific value of $r_n$. At a station where $r_n$ has always values less that 5% can shallow water effects always be neglected? events with "high" (beyond which threshold?) values of $r_n$ are poorly reproduced in the ERA-Interim reanalysis? these are examples of relevant questions, in my view.

I suggest that the authors make more clear what are the practical implications of their results and whether they can offer guidelines for the interpretation of existing data and model simulations, e.g. in terms of accuracy of results, of the model setup and characteristics to be used in the different areas, on the necessity to account for wave-current interaction.

The title does not really reflect the areas effectively included in the study. In depth analysis is concentrated in the North Sea and the Celtic Sea. Very little information is delivered for the rest of the European seaa, including shallow parts of the Mediterranean (Rhone Delta and north Adriatic), the Bay of Biscay and Baltic and Barents seas.
* * *

---

## Referee Comment (RC2) · Anonymous Referee #2 · 2 Mar 2017

**The "shallow-waterness" of the wave climate in European coastal regions**

*by Kai Håkon Christensen, Ana Carrasco, Jean-Raymond Bidlot, and Øyvind Breivik*

This short MS considers the extent to which waves around European coastal regions can be considered as 'shallow-water' i.e. where the wave circulation reaches to and is influenced by the sea-floor rather than 'deep-water' i.e. the vertical scale of the wave circulation is smaller than the sea depth. The approach taken here is to analyze waves in the ECMWF ERA-Interim coupled atmosphere-wave-ocean reanalysis. The criterion for 'shallow-waterness' is taken to be $n = c_g / c_p$ , the ratio of the group velocity $c_g$ to the phase velocity $c_p$ ; which is 0.5 for extreme deep-water waves and 1 for extreme, non-dispersive, shallow water waves where the ocean depth is much less than the vertical scale of decay $k^{-1}$ of the orbital motion, where $k$ is the wavenumber of the wave. For a given value of $n = n_c$ the key diagnostic is then the fraction of the wave energy in the part of the wave spectrum with $c_g / c_p \geq n_c$ i.e. wavelengths longer than the wavelength which gives $c_g / c_p = n_c$ .

The authors then consider the field of this shallow-water energy fraction for four different values of $n_c$ ranging from 0.55 to 0.85. Values are largest in winter and even then generally low except in the North Sea (particularly the southern part) and parts of the Arctic and Mediterranean shelves. A clear seasonal cycle (low fraction in summer, higher fraction in winter) is evident in the simulations at six coastal stations. There is considerable atmospherically-driven high frequency temporal variability  in the shallow-water energy fractions, with occasional high values happening at times with a wide range of significant wave height.

*General comments*

The analysis set out here is very clear, and even though the MS is short and simple, the results are interesting. However, the authors need to be more explicit as to why the shallow-waterness is important, and what different values of $n_c$ imply. With the current MS, it is unclear why any of the four values of $n_c$ are important. Presumably the appropriate value of the shallow-waterness parameter $n_c$ depends on the application (bottom mixing, surge prediction etc) —this needs to be discussed. Also, the authors need to justify why they chose to couch

their cutoff criterion in terms of $n = c_g/c_p$ rather than something simpler e.g. the orbital motion at the bottom relative to that at the surface, or even simply $kh$.

*Recommendation*

The manuscript is fundamentally publishable, but requires more discussion of the implications of different values of $n_c$.

---

## Author Comment (AC1) · 31 Mar 2017

*In this paper authors identify the fraction of energy that is affected by interaction with the bottom. They found that it depends on the mean wavelength (I would say this is obvious), it can be large also far from the coast if the water is sufficiently shallow (which in practice is the case for the central areas of the North Sea). Even where water is deep (100m), shallow water effects can be occasionally present if waves are sufficiently long.*

*While the paper is well written, concise and methodologically clear (I mean that mathematical definition—see formulas 2 and 3—of $r_n$ is clear, I have some difficulty to identify the real utility of this study. In my view authors should explain the practical relevance of a specific value of $r_n$. At a station where $r_n$ has always values less that 5% can shallow water effects always be neglected? events with "high" (beyond which threshold?)*

[Figure]

*values of $r_n$ are poorly reproduced in the ERA-Interim reanalysis? these are examples of relevant questions, in my view.*

Good points, which are also made by Ref. #2. Our choice of the ratio (2) is, as we now say explicitly, mainly motivated by our interest in radiation stress theory, in which the ratio between the group and phase velocity occurs naturally. We have added a short discussion at the end of Sec. 2.1 demonstrating the use of the ratios $n$ and $r_n$ (Eq. 4).

*I suggest that the authors make more clear what are the practical implications of their results and whether they can offer guidelines for the interpretation of existing data and model simulations, e.g. in terms of accuracy of results, of the model setup and characteristics to be used in the different areas, on the necessity to account for wave-current interaction.*

The practical implications depend on what aspects of the "shallow-waterness" that are of interest. With (4) we provide one example relevant to radiation stresses. As we also mention in our conclusions (lines 145-150), all the necessary information to assess the value of (4) can be taken from plots like those in Figs. 5 and 6, hence allowing for a quick assessment of the relevance of the shallow water effects.

*The title does not really reflect the areas effectively included in the study. In depth analysis is concentrated in the North Sea and the Celtic Sea. Very little information is delivered for the rest of the European seas, including shallow parts of the Mediterranean (Rhone Delta and north Adriatic), the Bay of Biscay and Baltic and Barents seas.*

The ERA-I data set is unfortunately not the best for studying the details in all regions in Europe and, as we now indicate in the conclusions (lines 140-145), this study should be regarded more as a "proof of concept". The benefit of the stations chosen here is that they are distributed over a wide range of depths and that the verification statistics of ERA-I for these locations are good. These statistics have been added in Table 1.

---

## Author Comment (AC2) · 31 Mar 2017

*The analysis set out here is very clear, and even though the MS is short and simple, the results are interesting. However, the authors need to be more explicit as to why the shallow- waterness is important, and what different values of $n$ imply. With the current MS, it is un- clear why any of the four values of $n$ are important. Presumably the appropriate value of the shallow-waterness parameter $n$ depends on the application (bottom mixing, surge prediction etc)—this needs to be discussed.*

*Also, the authors need to justify why they chose to couch their cutoff criterion in terms of $n = c_g/c_p$ rather than something simpler e.g. the orbital motion at the bottom relative to that at the surface, or even simply $kh$.*

Please also refer to our replies to Ref. #1 above. We have added a paragraph at the end

of Sec. 2.1 providing an example on how this type of analysis can be used to assess the relevance of the "shallow-waterness" in the context of radiation stress theory, which also explains our choice of the cutoff criterion (2). The Ref. correctly points out that other criteria can be used, in particular if other aspects of the "shallow-waterness" are under study, and we now say so in Sec. 2.1 and the expanded conclusions.

---

## Author Response (AR2)

Dear Dr. Hoppema,

In this final revision we have added a few, more specific comments and suggestions based on the input we have received at meetings etc., see the next to last paragraph in the conclusions. We have also made the required changes on lines 34-36 and 38. Finally, we changed the depth at station 62069 in both text and figures. This was a simple formatting error on our part and the calculations, the results and the conclusions are all unchanged.

Best wishes,
Kai Christensen

[revised manuscript text omitted]